# Poeciliid male mate preference is influenced by female size but not by fecundity

Luis R. Arriaga and Ingo Schlupp

Department of Biology, University of Oklahoma, Norman, OK, USA

## ABSTRACT

While female mate preference is very well studied, male preference has only recently begun to receive significant attention. Its existence is found in numerous taxa, but empirical research has mostly been limited to a descriptive level and does not fully address the factors influencing its evolution. We attempted to address this issue using preference functions by comparing the strength of male preference for females of different sizes in nine populations of four poeciliid species. Due to environmental constraints (water toxicity and surface versus cave habitat), females from these populations vary in the degree to which their size is correlated to their fecundity. Hence, they vary in how their size signals their quality as mates. Since female size is strongly correlated with fecundity in this subfamily, males were sequentially presented with conspecific females of three different size categories and the strength of their preference for each was measured. Males preferred larger females in all populations, as predicted. However, the degree to which males preferred each size category, as measured by association time, was not correlated with its fecundity. In addition, cave males discriminated against smaller females more than surface males. Assuming that male preference is correlated with female fitness, these results suggest that factors other than fecundity have a strong influence on female fitness in these species.

## INTRODUCTION

The existence and evolution of female mate choice has received substantial attention in both theoretical and empirical grounds. However, male mate choice has been comparatively neglected until recently. This is because females typically have a larger *a priori* investment in any given mating event, and they are also inherently limited in the number of offspring they are able to produce (*Trivers, 1972*). The selective pressures giving rise to female mate choice are therefore obvious. However, while these pressures are often stronger in females, similar pressures are also experienced by males in many species. Males are limited in the proportion and quality of females they are able to fertilize, and can therefore maximize their fitness by selectively allocating their resources towards certain females. Theory thus predicts that male mate choice can be selected for under

Corresponding author
Luis R. Arriaga,
ouroboricalmus@yahoo.com

the following circumstances: (1) There is substantial male effort in terms of searching, courtship, mating, and mate guarding (*Pomiankowski, 1987*); (2) Females are scarce due to a biased operational sex ratio (*Van den Berghe & Warner, 1989*); (3) Female quality varies (*Johnstone, Reynolds & Deutsch, 1996*); (4) Males invest in parental care (*Sargent, Gross & van den Berghe, 1986*); and (5) Males' mating opportunities are limited and/or insemination success varies between different females (*Nakatsuru & Kramer, 1982*; *Verrell, 1985*; also see reviews of *Bonduriansky, 2001*; *Edward & Chapman, 2011*).

There is some evidence that these factors have indeed resulted in male preference in species ranging from sexually cannibalistic arthropods, fish and birds with heavy parental investment, and polygynous species without parental care (see reviews by *Amundsen, 2000*; *Bonduriansky, 2001*; *Edward & Chapman, 2011*). However, it's difficult to determine the specific factors driving the evolution of male choice in these systems since multiple factors predicted to drive male mate choice evolution are present in these species. Previous empirical research has often been limited to demonstrating the existence of male mate choice and describing its manifestation in particular species. We are not aware of research examining the evolution of the strength of male preference in response to specific selective pressures. We attempted to address this by comparing poeciliid populations in which variation in female quality is likely to be the main driving force behind male choice evolution.

Poeciliids are a family of internally-fertilizing, promiscuous fish that form mixed-sex shoals and give birth to live young. Previous studies have demonstrated male preference for larger females in many species (*Abrahams, 1993*; *Bisazza, Marconato & Marin, 1989*; *Dosen & Montgomerie, 2004a*; *Gumm & Gabor, 2005*; *Herdman, Kelly & Godin, 2004*; *Hoysak & Godin, 2007*; *Jeswiet & Godin, 2011*; *Plath et al., 2006*; *Ptacek & Travis, 1997*). This is likely a result of the strong correlation between female size and fecundity (*Herdman, Kelly & Godin, 2004*; *Hughes, 1985*; *Marsh-Matthews et al., 2005*; *Milton & Arthington, 1983*; *Reznick & Endler, 1982*; *Riesch et al., 2009b*), suggesting that size is used as a signal of female quality and has played a role in the evolution of male mate choice.

While previous studies have shown that males prefer more fecund females, how the strength of this preference changes as female fecundity evolves has not been investigated. To address this, we selected a number of populations from four poeciliid species (*Poecilia mexicana, Limia sulphurophila, Gambusia sexradiata*, and *Gambusia eurystoma*) that exhibit different relationships between female size and fecundity. These different relationships evolved as a response to living in different habitats (Fig. 1), and have been found to persist even in fish raised in common garden conditions (*Riesch et al., 2009b*; R Riesch et al., unpublished data). Living in a toxic habitat or living in a cave independently led to larger and fewer offspring; in other words, larger and fewer offspring are found in toxic habitats (whether on the surface or in a cave), as well as cave habitats (whether toxic or nontoxic); smaller and more numerous offspring are found in nontoxic surface habitats (*Riesch et al., 2009b*; *Riesch et al., 2010b*; *Riesch, Plath & Schlupp, 2010*). Because female size and fecundity decoupled from each other in this system, it is possible to comparatively determine how female fecundity affects the evolution of male preference.

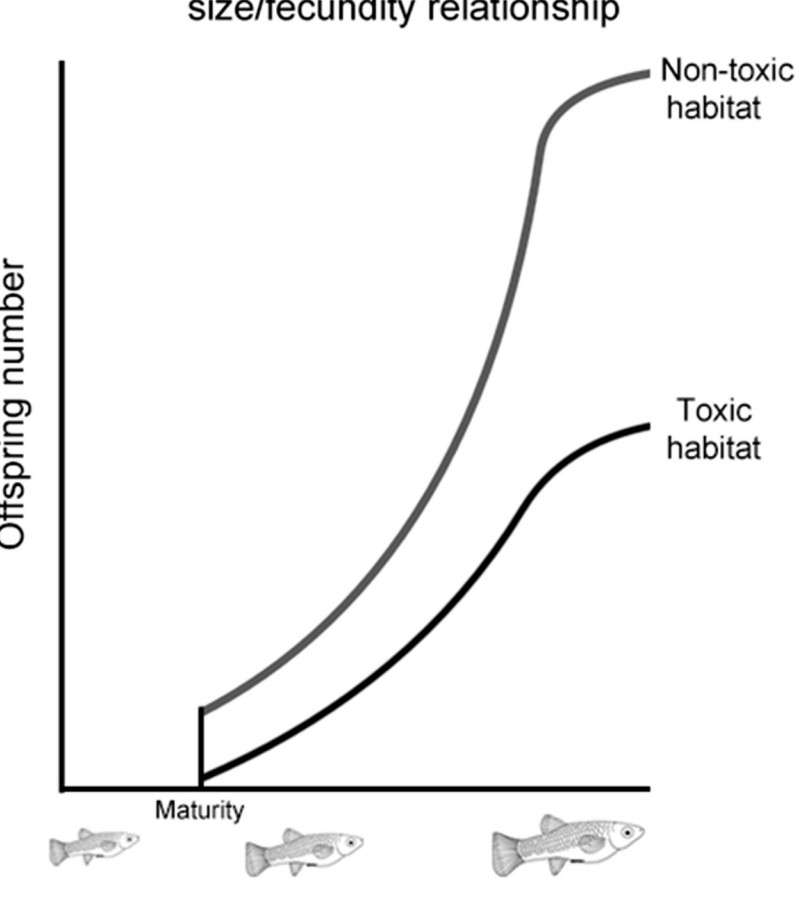

**Figure 1** **Effect of habitat toxicity on the size/fecundity relationship.** Schematic illustration of the previously-established relationship between female poeciliid size and her fecundity in toxic and non-toxic habitats.

Mate preference is most commonly studied using dichotomous choice tests, where focal individuals are given a choice between two stimuli (*Ritchie, 1996*). While this is a powerful approach to assess mate preferences within populations, this approach makes it difficult to compare between populations. Absolute preference functions are an alternative method that allows the preferences between populations to be compared (*Wagner, 1998*). Absolute preference functions measure the preference of individual males for females varying in a continuous trait. This is done by sequentially presenting individual females to each male, allowing the shape of a male's preference for that trait to be quantified. The resulting correlation can thus be thought as being the probability that a given male will accept a particular female trait (*Ritchie, 1996*). Such association preferences are commonly used to study male mating preferences in poeciliids and have been shown to correlate well with actual mating choices (*Dosen & Montgomerie, 2004b*; *Plath et al., 2006*; *Schlupp & Ryan, 1997*; *Wong, Fisher & Rosenthal, 2005*).

**Table 1 Populations used.** Collection details and habitat characteristics of the populations from which the individuals used originated. All individuals were descendants of these original populations and were raised in common garden conditions.

| Population | Source location | | Year | Toxic/non-toxic | Cave/surface |
|---|---|---|---|---|---|
| *P. mexicana*-Oxolotan | 17° 26′ 55″ N | 92° 45′ 55″ W | 2005 | Non-toxic | Surface |
| *P. mexicana*-PS0 | 17° 26′ 30″ N | 92° 46′ 30″ W | 2005 | Toxic | Surface |
| *P. mexicana*-Luna | 17° 26′ 35″ N | 92° 46′ 39″ W | 2006 | Non-toxic | Cave |
| *P. mexicana*-PSV | 17° 26′ 30″ N | 92° 46′ 30″ W | 2005 | Toxic | Cave |
| *P. mexicana*-PSX | 17° 26′ 30″ N | 92° 46′ 30″ W | 2005 | Toxic | Cave |
| *P. mexicana*-PSXIII | 17° 26′ 30″ N | 92° 46′ 30″ W | 2005 | Toxic | Cave |
| *G. eurystoma* | 17° 33′ 10″ N | 92° 59′ 51″ W | 2006 | Toxic | Surface |
| *G. sexradiata* | 17° 59′ 56″ N | 93° 8′ 11″ W | 2006 | Non-toxic | Surface |
| *L. sulphurophila* | 18° 23′ 52″ N | 71° 34′ 12″ W | 2006 | Toxic | Surface |

The present study had two goals: (1) to test whether male preferences can be detected using preference functions, and (2) to see if male preference tracks changes in female fecundity in these populations. Our prediction was that male preference for larger females would be stronger in populations from nontoxic environments, where the relative increase in female fecundity is higher as compared to populations from toxic environments.

## MATERIALS & METHODS

### Species and populations

Nine populations of four poeciliid species representing different habitat types were used (summarized in Table 1). *Gambusia eurystoma* is a surface species endemic to the sulfidic Baños del Azufre in Tabasco, Mexico (*Tobler et al., 2008*). *Limia sulphurophila* is another surface fish living in a sulfidic habitat, but it is endemic to a small pool in the island of Hispaniola (*Rivas, 1980*; *Rivas, 1984*). The population of *G. sexradiata* used lives in non-sulfidic surface waters. The six populations of *P. mexicana* used in this study live in different habitats. The Oxolotan population is named after the non-sulfidic, surface river it originates from. The PS0 population also lives in a surface creek, but whose water is sulfidic. The water from this creek, named El Azufre, originates from Cueva del Azufre, a sulfidic cave from which three of the other populations originated. These populations, inhabiting a dark and toxic environment, are the PSV, PSX, and PSXIII populations (for a schematic map of the region and of the cave, see *Plath et al., 2010*; *Tobler et al., 2006*). They are named after the chamber of the cave in which they live. The sixth population is the Luna population, which originates from a non-toxic cave of the same name (*Tobler et al., 2008*).

Fish from all of the populations are maintained in flow-through stock tanks in the Aquatic Research Facility at the University of Oklahoma, and have been in captivity for varying lengths of time (Table 1). These tanks were the immediate source of the fish used in this study. All are maintained in nonsulfidic, common garden conditions inside a greenhouse that receives natural light.

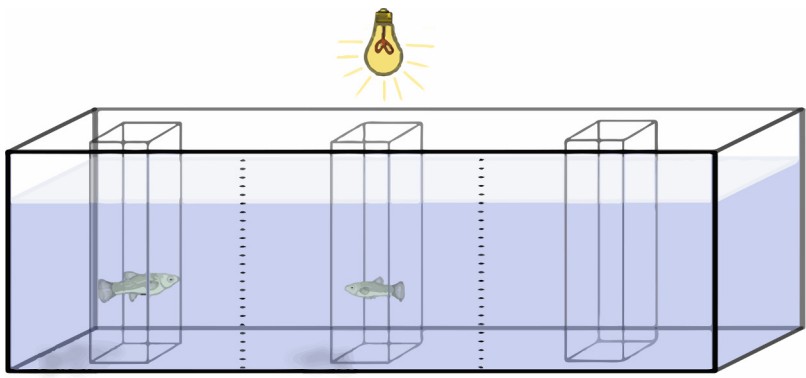

**Figure 2 Experimental setup.** Schematic representation of the experimental setup during acclimation period. Gravel and cylinder perforations were omitted for clarity.

Fish from the stock tanks were caught with a small seine and segregated by sex. Mature females were then selected using minimum standard length (tip of the snout to the end of the vertebral column) as the criterion for sexual maturity. Due to natural size differences between species, the exact criterion used varied (*P. mexicana:* 29 mm for the Luna population, and 30 mm for all others; *G. eurystoma:* 22 mm; *G. sexradiata*: 18 mm; and *L. sulphurophila*: 21 mm). The mature females were then sorted into roughly equally sized groups according to size (small, medium, and large), and were then placed in different 38 L stock tanks. Males were randomly assigned an ID number that determined the order in which they would be tested, and they were housed in individual 5 L tanks that were out of sight of the females.

## Experimental setup

Preference functions are established by measuring the amount of time a focus individual spends in association with different stimulus fish. These stimulus fish are presented sequentially and differ in the variable in question. In this case, females of different sizes were sequentially presented to a male, and the time that the male spent with each female was recorded. To do this, a 76 L aquarium, with gravel spread evenly to reduce potential bias from fish being distracted by uneven gravel, was divided lengthwise into three equal sections with two vertical lines drawn on the glass (Fig. 2). The outer two sections were considered the "preference zones", while the central section was considered a "neutral zone".

Three hollow square prisms (or "cylinder") made out of clear plexiglass were located in the center of each section of the tank. The cylinder in the center of the tank had solid walls, while the two outer cylinders were perforated with seven circular holes 6 mm in diameter to allow for chemical and mechanosensory signals. Chemical and mechanosensory signals have been found to be important factors in poeciliid mating behavior, influencing the repeatability of individual preferences as well as the overall preference (*Coleman, 2011*; *Hoysak & Godin, 2007*; *Plath et al., 2006*; *Rüschenbaum & Schlupp, 2012*). All three cylinders were 8.5 cm long by 8.5 cm wide, and tall enough to stick through the water. To reduce visual distractions, three sides of the tank were covered. The observer sat in a

chair 2 m from the tank and observed the fish through the front pane of the tank. A light with a 60 W, "soft white" light bulb was placed 30 cm above the center of the tank for illumination.

### Testing procedure

A randomly selected female from the predetermined size category (small, medium, or large) was placed in the cylinder in one of the outer preference zones. The order of the females which each male would be presented with, as well as the side in which each female would be placed, were randomly determined. A male was then placed in the central cylinder for 5 min. After the 5 min of acclimation, the cylinder around the male was gently removed. Using two stopwatches, the amount of time the male spent in the preference zone containing the female was measured by the observer. This was done for 5 min, after which the fish were removed from the tank. The water was then stirred to homogenize any lingering chemical signals from affecting the results of future trials. Another pair of fish was then placed in their corresponding cylinders to acclimate. Every three pairs, a partial water change was also made previous to the acclimation period of the next fish. Male weight and standard length were also measured and used as covariates, but were not included in the final model because neither was significant.

### Statistical analysis

After checking the assumptions, a mixed between-within subjects ANOVA was performed to determine the effect of two habitat variables on male preference for small, medium, and large females. The two habitat variables used were "cave habitat", whether the population originated from a cave or from a surface stream, and "toxicity", whether the population originated from a toxic or non-toxic stream. Because the raw results were not normally distributed, the male preference variables were reflected and square root transformed to meet the normality assumption.

Experiments were approved by the University of Oklahoma IACUC (R09-030).

### RESULTS

All statistical assumptions were met after the data transformation, with the exception of homogeneity of variances for the time males spent with medium females ($p = 0.026$). However, this violation was not deemed to be severe enough to invalidate the ANOVA. As expected, there was a significant main effect for time spent with larger females, regardless of the habitat of origin (Wilks' Lambda $= 0.911$, $F_{(2,123)} = 5.99$, $p = 0.003$, $h_p^2 = 0.089$). The main between-subjects effect comparing toxicity was not significant ($F_{(1,124)} = 0.047$, $p = 0.829$, $h_p^2 = 0.000$), nor was the main effect for cave habitat ($F_{(1,124)} = 1.043$, $p = 0.309$, $h_p^2 = 0.008$), or the interaction between cave habitat and toxicity ($F_{(1,124)} = 0.011$, $p = 0.917$, $h_p^2 = 0.000$). These results suggest that, correcting for the effect of female size, habitat type does not affect male preference values.

There was also a significant interaction between cave habitat and the time males spent with females from different size categories (Wilks' Lambda $= 0.934$, $F_{(2,123)} = 4.36$, $p = 0.015$, $h_p^2 = 0.066$; Fig. 3). However, there was no significant interaction between the

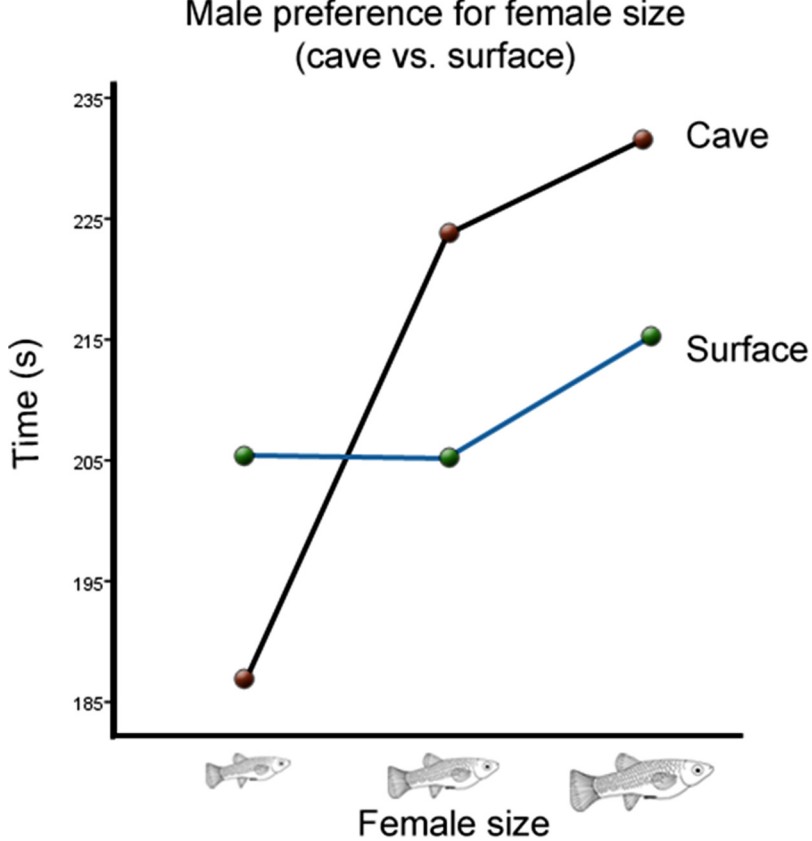

**Figure 3 Male preference for female size (cave vs. surface).** Average transformed male preference for female size in cave vs. surface habitats. There is a significant difference between the two preference functions.

time males spent with females of different sizes and the toxicity of the habitat from which they originated (Wilks' Lambda $= 0.996$, $F_{(2,123)} = 0.26$, $p = 0.77$, $h_p^2 = 0.004$; Fig. 4). There was also no significant interaction between time, cave habitat, and toxicity together (Wilks' Lambda $= 0.956$, $F_{(2,123)} = 8.86$, $p = 0.06$, $h_p^2 = 0.044$). These results suggest that the presence or absence of hydrogen sulfide in the population's habitat of origin does not influence males' preference for female size, but the cave habitat does. Descriptive statistics are summarized in Table 2.

## DISCUSSION

As predicted, males did exhibit a general preference for larger females when all populations were considered in aggregate. This result is consistent with previous dichotomous-test studies finding preference for larger females in poeciliids (*Bisazza, Marconato & Marin, 1989*; *Herdman, Kelly & Godin, 2004*; *Hoysak & Godin, 2007*; *Jeswiet & Godin, 2011*; *Plath et al., 2006*) and indicates that absolute preference functions are an accurate tool to study individual preferences. Because preference functions can be used to compare preferences between individuals, they can also be used to address a more specific and broader range of questions than is possible using only dichotomous choice tests.

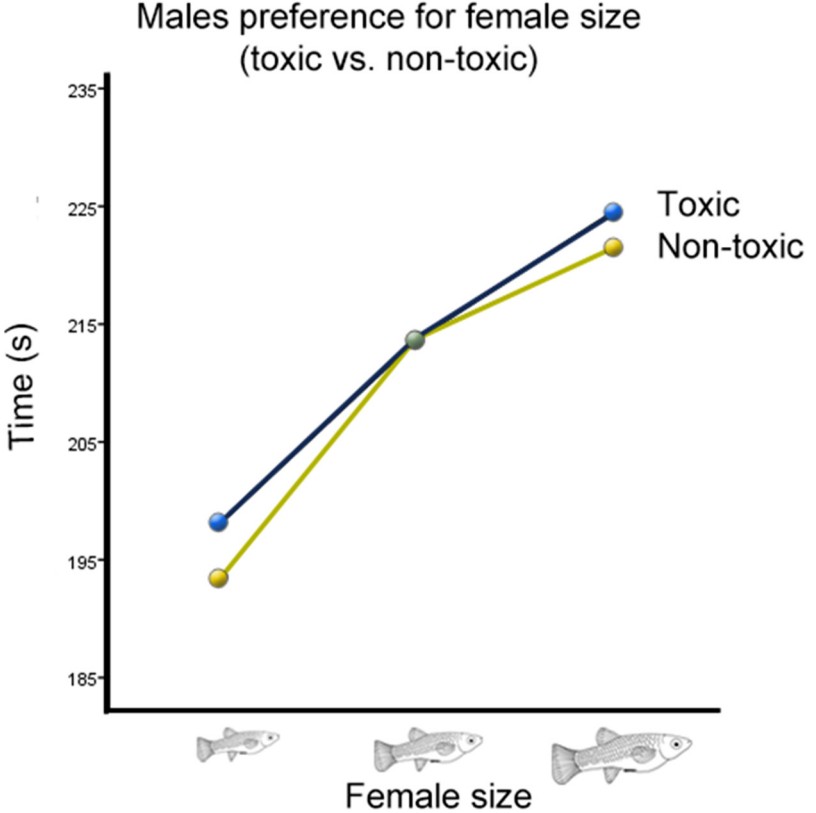

**Figure 4  Male preference for female size (toxic vs. non-toxic).** Average transformed male preference for female size in toxic vs. benign habitats. Preference functions are not significantly different from each other.

**Table 2  Descriptive statistics.** Sample size, average time, and standard deviation that males from each of the populations spent with each female size category.

|  |  | N | Mean association time (s) | Std. deviation |
|---|---|---|---|---|
| Small female | Cave | 56 | 191.4 | 74.9 |
|  | Surface | 72 | 205.1 | 59.1 |
|  | Nontoxic | 43 | 199.5 | 75.7 |
|  | Toxic | 85 | 198.8 | 61.9 |
| Medium female | Cave | 56 | 218.1 | 59.4 |
|  | Surface | 72 | 206.9 | 50.9 |
|  | Nontoxic | 43 | 206.8 | 62.4 |
|  | Toxic | 85 | 214.3 | 50.8 |
| Large female | Cave | 56 | 233.2 | 62.2 |
|  | Surface | 72 | 215.1 | 62.2 |
|  | Nontoxic | 43 | 219.1 | 56.4 |
|  | Toxic | 85 | 225.0 | 65.7 |

**PeerJ** ___________________________________________________

In addition to an overall preference for larger females, the strength of male preference should reflect how tightly correlated female size is with female fecundity. The populations tested originated from habitats with different combinations of two variables—water toxicity as a result of the presence or absence of $H_2S$ ("toxicity"), and epigean or hypogean habitat ("cave habitat"). Previous studies have shown that in *P. mexicana* (*Riesch et al., 2009b*; *Riesch, Plath & Schlupp, 2010*), as well as in *G. sexradiata* and *G. eurystoma* (*Riesch et al., 2010b*), female fecundity is strongly correlated with toxicity. Females from toxic habitats have much larger, but fewer, offspring. Because of this, there is a larger change in fecundity from small to large females in nontoxic habitats. The main hypothesis was therefore that the preference function of males from toxic habitats would be less steep than that of males from nontoxic habitats. The results did not support our hypothesis, as there was no significant interaction between time spent with a female and the toxicity of the habitat the male originated from.

This result suggests that the change in female fecundity experienced from benign to toxic habitats is only weakly correlated with the change female quality. The reason for this is unclear and could be due to a combination of factors. It is possible that female size in nontoxic habitats is correlated with increased female mortality and/or with a decrease in offspring quality. An alternative possibility is that female size and quality are highly correlated, but that there has been insufficient time for male preference to change as a response to female adaptation. It is currently unknown how long the populations in toxic habitats have been adapting to their environments, and how recently the changes in female fecundity have evolved. Since male preference is likely under weaker selection than other traits, it is possible that males from toxic populations have either not had a sufficient amount of time to adapt, or that the amount of gene flow from nontoxic populations has been able to counteract the effects of selection. It is currently unknown how much gene flow there is between toxic (*P. mexicana*: PS0, *G. sexradiata*: populations not used in the present study) and nontoxic (*P. mexicana*: Oxolotan, *G. sexradiata*) surface populations.

While the amount of gene flow between toxic and non-toxic populations still needs to be determined, it is known that there is very little gene flow between *P. mexicana* populations from the Cueva del Azufre (PSV, PSX, and PSXIII) and those in the surface (*Plath et al., 2010*). The genetic isolation of cave fish from surface fish might be an important reason why cave habitat did have a significant effect on male preference. This difference seems to be mainly derived from cave males' relatively low preference for small females (Fig. 3), as the change in preference for medium to large females is nearly identical in males from both habitats. This suggests that males are not responding to fecundity or offspring size per se, since one would expect that the change of preference from medium to large females would differ between the two environments. This same logic also suggests that the difference is not due to cave males being choosier than surface males. If males preferred larger females due to the cave habitat leading to a greater cost in male effort, males would disproportionally prefer large females over medium females as well. Instead, these results indicate that there is a relative disadvantage for cave males to mate with smaller females, which could result if small cave females have lower fitness than small surface females.

A fitness difference could occur if mortality rates for small cave female are higher than those of medium or large females, perhaps as a result of differential predation pressures. No direct evidence exists on the relative predation pressures between the two habitats, but there is reason to believe that this possibility might be true. Cave populations of *P. mexicana* are known to be preyed upon by predators which hunt by sensing tactile and/or chemical signals from fish in close proximity (*Horstkotte et al., 2010*; *Tobler, Franssen & Plath, 2008*; *Tobler et al., 2009*), and may prey disproportionally upon smaller females. At the same time, surface populations experience very different predation pressures: surface fish are preyed upon by large visual predators (*Riesch et al., 2009a*; *Riesch et al., 2010a*; *Tobler et al., 2006*) that target larger fish (*Trexler, Tempe & Travis, 1994*). Thus, medium and large surface females are likely to experience greater predation pressure than small females.

In summary, we have shown that (1) male preference for larger female size exists. This is consistent with previous research, indicating that absolute preference functions are a valid approach in this system. (2) Hydrogen sulphide does not affect the shape of male preference function for female size in this system. Since $H_2S$ greatly affects female fecundity, this suggests that inter-population differences of fecundity are not very highly correlated with inter-population differences in fitness. Alternatively, there has not been enough time or enough selection pressure to allow male preference to evolve as a response to changes in female fecundity, or gene flow has been large enough to negate the effects of these pressures. (3) Cave habitat, independent of water toxicity, does affect male preference. Cave males had a relative lack of preference for small females. We suggest that this could be a result of differences in predation pressure which could lead to relatively increased mortality for small females in the caves, relatively increased mortality for medium and large females in the surface, or both. If true, this would highlight the role that predators play in the evolution of male mate choice.

## ACKNOWLEDGEMENTS

We thank Millard L. Henry for his help with the experiments, and Sam B. Rhodes and Amber M. Makowicz for their help with fish care. We are grateful to Dr. Constantino Macías Garcia and an anonymous reviewer for comments on a previous version of the manuscript.

### Funding

IS gratefully acknowledges support from the Alexander von Humboldt Foundation. The funders had no role in study design, data collection and analysis, decision to publish, or preparation of the manuscript.

## Grant Disclosures

The following grant information was disclosed by the authors:
Alexander von Humboldt Foundation.

## Competing Interests

The authors declare no competing interests.

## Author Contributions

- Luis R. Arriaga conceived and designed the experiments, performed the experiments, analyzed the data, wrote the paper.
- Ingo Schlupp conceived and designed the experiments, contributed reagents/materials/analysis tools, wrote the paper.

## Animal Ethics

The following information was supplied relating to ethical approvals (i.e., approving body and any reference numbers):

Experiments were approved by the University of Oklahoma IACUC (R09-030).

## Supplemental Information

Supplemental information for this article can be found online at http://dx.doi.org/10.7717/peerj.140.

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
