# Peer review of "Poeciliid male mate preference is influenced by female size but not by fecundity"

_PeerJ, doi:10.7717/peerj.140_

## Round 0.1 · original submission · Major Revisions

I found this ms to address and interesting question, and to be generally thorough, as did the reviewers. Also like the reviewers I found the use of preference functions to be a nice addition to the ms. The reviewers pointed out a few additional analyses (described below) that they felt it was necessary to explore before publication. Please follow or address these suggestions.

·

Basic reporting

This is a competent report on an creative experiment to evaluate the possibility that (the intensity of) male mate choice varies between species and populations whose females differ in their pattern of investment on offspring as a consequence of inhabiting contrasting habitats (toxic vs non-toxic, cave vs surface dwelling). The rational places the work within the relevant conceptual framework and the introduction provides much useful background information (with one caveat; see below). The authors suggest that this may be the first evaluation of variation in male mate choice prompted by ecological differences between species/populations. I believe that Kelley et al (Nature; doi:10.1038/44314) demonstrated local differences in male choosiness (whether they mate with familiar females or not) which are consequence of ecological factors (which determine the size of the shoals). The authors describe the pattern -common to many viviparous fish- of male preferences or large females, which is the consequence of female fecundity being a function of female size. They further indicate that females from toxic environments produce fewer, but larger offspring that females elsewhere. This they construe (and this is my caveat) as meaning that a given difference in female size should reflect a different change in fecundity for female from toxic locations than for females from non-toxic places. Indeed, if fecundity is taken to merely imply number of fish, and not their quality (in the context of their population), this may be an adequate inference. yet form the point of view of the males, the relative value of tow females differing in size would depend on the of the regression between female size and her fecundity relative to that of the population (i.e as a proportion of the maximum fecundity achieved by local females). I think the authors should show the slopes of such regression lines and demonstrate that they are indeed steeper in toxic localities than elsewhere. I suspect they are not (that they are only less smooth -made of larger steps-), and this may explain their negative results but does not invalidate the study.

Experimental design

Given the rationale of the study, the sample of species/populations and the procedures to evaluate male mating preferences are appropriate. I particularly applaud the fact that the authors characterize preference curves, rather than merely evaluating preferences in dichotomous tests. This has only a few times being done in fish (and indeed in general), and only when dealing with (fish) female mating preferences. The result are, thus, sound and publishable, even if they do not prove the research hypothesis (but see my comments above).

Validity of the findings

There is one aspect of the analysis that I am concerned with. The authors included male size as a co-variate and then excluded it from the final model once it turned not to have a significant effect on the response variable. Yet we are informed that female size (at maturity at least) differs between localities, thus male size should be adjusted (expressed as a proportion of female size), as female size is also made relative (i.e. a "large" female from one population may be very different in size from a "large" female from another). Still, my more important suggestion would be that the slopes of fecundity (relative to the local population) on female size are calculated and compared with the slopes (differences) of male preference as a function of female size.

Additional comments

I think the discussion of the negative results would be much more lean and fruitful if the assumption that differences in the pattern of maternal allocation (few large vs many small embryos) determines differences in the slope of relative fecundity on female size (and not only in the smoothness of those lines) was formally evaluated.

Reviewer 2 ·

Basic reporting

It was not clearly shown that size is a poorer indicator of fecundity within the toxic environment (Figure 1, which does not appear to be real data either), or in the cave versus open water environments. I have made some suggestions of how they might want to do this below. I think it is important to include this data in the current study.

Experimental design

No comment

Validity of the findings

In addition all the fish from different populations are analyzed together with no consideration of their locality or species identify. This adds some independence issues to the analyses. One possibility would be to use a mixed effects model, where this information could be included as a random factor.

Finally, the manuscript does not do a very good job at explaining the results in relation to cave habitat. There was an interaction between female size and habitat that explained variation in preference for female size. The authors suggest that males from the cave populations “did not prefer” smaller females, but this needs to be discussed in the context of a preference function, not a preference for a particular size class, or relative to the males from the other habitat (surface) that spent more time with the smaller females.

Additional comments

Overall, it is difficult to know what to take away from the study. The last sentence of the abstract makes a statement about female fitness and fecundity, even though the paper did not examine female fitness but was asking how female fecundity influences male mate preference.

A few more specific comments:

Line 40-44. Statement that because female size is correlated with fecundity, that size is used as a signal of female “quality” makes it sound like you are suggesting that fecundity is the same thing as female quality?

Line 45. The previous studies found a preference for larger female size, and assumed it was due to fecundity. Did they demonstrate larger females were more fecund females? This is relevant to your study

Line 49. How do you know these relationships between fecundity and female size have “evolved” and are not just environmentally plastic responses to toxic environments? I don’t think they have to have evolved for what you are interested in determining here.

Line 50. Larger and fewer offspring does not necessarily mean there is no relationship between female size and fecundity. This is an important point, as the males living in a toxic environment are not comparing females from this environment to those from a nontoxic (more and smaller fry). As shown in figure 1, there is a relationship between female size and fecundity in both environments, and for most of the distribution the slope is appears to be the same. I think what would be more telling, is the variation around this relationship, producing either 1) different slopes or 2) lower r2. Also, why is Figure 1 a “schematic” and not real data if these relationships have been established previously?

Line 67. I think you mean mate “choices” not “preferences” again here.

Line 70. Mate choice will be more “acute”? Do you mean stronger preference for female size?

Line 159. Interaction between habitat and time male spent with females? You are trying to explain variation in time spent with females, so the interaction has to be between habitat and something else. Female size, yes? Same with line 161.

---

## Round 0.2 · accepted · Accept

The authors have provided what I view as convincing responses to the reviewer's comments. It would perhaps have been more helpful if they had also added some discussion of these points to the ms to provide clarity for future readers, so they may want to consider still doing this.